# Upregulation of long non-coding RNA *ROR1-AS1* promotes cell growth and migration in bladder cancer by regulation of miR-504

**Qingke Chen**[1]*, **Lingmin Fu**[2]

**1** Department of Urology, The First Affiliated Hospital of Nanchang University, Nanchang, China, **2** Jiangxi Health Vocational College, Nanchang, China

* cqkurethral@126.com

## Abstract

### Background

Increasing evidence has suggested that multiple long non-coding RNAs (lncRNAs) act key regulatory functions in the pathogenesis of bladder cancer. This study aimed to determine the expression and clinical significance of lncRNA ROR1 antisense RNA 1 (*ROR1-AS1*) from patients with bladder cancer, and to explore the potential role and mechanism underlying ROR1-AS1-related cancer progression.

### Methods

Real time quantitative PCR (RT-qPCR) was conducted to detected the expression levels of *ROR1-AS1* and miR-504 in bladder cancer samples and cell lines. Chi-square test was used for correlation analysis. 3-(4,5-dimethyl-2-thiazolyl)-2,5-diphenyl-2-H-tetrazolium bromide (MTT) and wound scratch assays were applied to assesses the effects of *ROR1-AS1* overexpression and knockdown on bladder cancer cell growth and migration *in vitro*, respectively. The prognosis of bladder cancer patients was evaluated by survival curves with Kaplan-Meier method. The regulatory mechanism of *ROR1-AS1* on miR-504 was confirmed by bioinformatics analysis and luciferase reporter gene assay.

### Results

*ROR1-AS1* levels were obviously upregulated in bladder cancer tissues than matched normal bladder tissues. High expression of *ROR1-AS1* was remarkably correlated with higher histological grade, advanced tumor stage, and positive lymph node metastasis. High *ROR1-AS1* expression was markedly correlated with shorter overall survival of bladder cancer patients. Moreover, knockdown of *ROR1-AS1* notably repressed T24 and 5637 cell growth and migration. *ROR1-AS1* directly bound with miR-504 and act as a molecular sponge to decrease miR-504 expression. Silencing of miR-504 partly abrogated *ROR1-AS1* knockdown-induced inhibitory effects on bladder cancer cell growth and migration.

**Data Availability Statement:** All relevant data are within the manuscript.

**Funding:** The authors received no specific funding for this work.

**Competing interests:** The authors declare that they have no competing interests.

**Abbreviations:** *GAPDH*, glyceraldehyde-3-phosphate dehydrogenase; LncRNAs, long non-coding RNAs; MicroRNAs, miRs; MTT, 3-(4,5-dimethyl-2-thiazolyl)-2,5-diphenyl-2-H-tetrazolium bromide; *ROR1-AS1*, ROR1 antisense RNA 1; RT-qPCR, real time quantitative PCR.

## Conclusions

Our data demonstrated that increased *ROR1-AS1* promotes cell growth and migration of bladder cancer via regulation of miR-504, indicating *ROR1-AS1* may be used as a prognostic biomarker and therapeutic target for bladder cancer.

## Introduction

Bladder cancer is the fourth most common diagnosed malignancy, and is one of the most expensive malignancies to treatment in men worldwide [1]. According to the Global Cancer Statistics in 2017, approximately 440,000 new cases are diagnosed with bladder cancer, and among those patients about 130,000 cases died of cancer [2]. Patients with invasive bladder cancer are commonly treated with radical cystectomy and urinary diversion; however, those patients usually have a poor prognosis [3,4]. At present, because of the poor understanding of pathological mechanisms in the progression of bladder cancer, the effective treatment for this cancer is very limited [5,6]. Therefore, it's essential to identify new sensitive and promising therapeutic targets for bladder cancer.

Long non-coding RNA (lncRNA) is a class of non-coding RNA which surpasses 200 nucleotides in length, usually exhibiting little or no coding potential [7]. Increasing data have demonstrated that lncRNAs play a crucial role in modulating various cellular biological processes, including growth and differentiation, apoptosis, malignant metastasis, and epithelial-mesenchymal transition [8–10]. According to their location on the human genome, lncRNAs can be placed into five broad categories: sense, antisense, bidirectional, intronic, and intergenic. Recent studies have identified that some antisense lncRNAs participate in the tumorigenesis and carcinogenesis of diverse human cancers, these lncRNAs can also be used as potential and effective biomarkers for cancer therapies [11,12]. For instance, ABHD11 antisense RNA1 acts as an oncogene and a potential target for antitumor therapies in ovarian cancer [13]. Long intergenic non-protein coding RNA 1133 inhibits breast cancer cell invasion and metastasis by negatively regulating SRY-box transcription factor 4 expression via enhancer of zeste 2 polycomb repressive complex 2 subunit [14]. SMAD5 antisense RNA 1 functions as a miR-106a-5p sponge to promote epithelial mesenchymal transition in nasopharyngeal carcinoma [15]. Additional, DLX6 antisense RNA 1 promotes cell growth and invasiveness in bladder cancer via modulating the miR-223-HSP90B1 axis [16].

ROR1 antisense RNA 1 (*ROR1-AS1*) is a novel found lncRNA, which locates at human genome 1p31.3 position. Hu and his team [17] in 2017 reported that *ROR-AS1* is involved in regulation of gene transcription via associating with polycomb repressive complex 2 complex, and may serve as a new biomarker in patients with mantle cell lymphoma. Recently, several studies indicated that *ROR1-AS1* can enhance colorectal cancer metastasis and tumorigenesis [18,19]. MicroRNAs (miRs) act as tumor suppressive or oncogenic factors and are major post-transcriptional gene regulators in diverse cancers. LncRNAs can associate with miRs to impact cell biological behaviors by acting as ceRNAs by competitively binding common miRs [19]. Despite the above-mentioned knowledge, how *ROR1-AS1* participated in the progression of bladder cancer remains unclear. The present study aimed to explore the functional role and downstream target miRs of *ROR1-AS1* in the progression of bladder cancer. We firstly analyzed the expression and clinical significances of *ROR1-AS1* in patients with bladder cancer. Then, we performed *in vitro* loss-of-function and gain-of-function experiments to determine the potential function of ROR1-AS1 in bladder cancer cell growth and migration. Finally,

luciferase reporter gene assay and rescue experiments were applied to confirm the regulatory mechanism of *ROR1-AS1* on miR-504. Our study highlighted a key role of ROR1-AS1/miR-504 axis in the progression of bladder cancer.

## Materials and methods

### Clinical specimens and cell culture

This research was approved by the Ethics Committee of the First Affiliated Hospital of Nanchang University (Approval No. 2018070), and the written informed consents were received from all patients in accordance with the 1964 Helsinki declaration and its later amendments. This retrospective study included 65 cases of bladder cancer patients who underwent surgery at Department of Urology, First Affiliated Hospital of Nanchang University between 09/2011 and 05/2017. Human bladder tissues and adjacent matched normal bladder tissues ($\geq$ 3 cm away from the edge of cancer) from all patients were collected between 09/2011 and 05/2017, and instantly frozen in liquid nitrogen after operation until further use. The clinicopatholigcal information were recorded and summarized by patient's attending physician. The grade of bladder cancer was assessed according to the World Health Organization 2004 Grading System [20] and the stage was classified according to the modified tumor–node–metastasis (TNM) cancer staging system (UICC, 2002) [21]. The research was carried out at First Affiliated Hospital of Nanchang University during 01/2018-08/2019.

Five human bladder cancer cells (T24, 5637, J82, 253J and RT4) and a normal bladder epithelial cell (SV-HUC-1) were directly purchased from the American Type Culture Collection (ATCC, Manassas, VA, USA) and maintained in DMEM medium (Thermo Fisher Scientific, Waltham, MA, USA) and 10% FBS (HyClone, Logan, UT, USA) with an incubator containing 5% $CO_2$ at 37°C. The SV-HUC-1 cell were cultured in F-12K medium (Thermo Fisher Scientific) plus 10% FBS and antibiotics (100 U/ml penicillin and 100 μg/ml streptomycin).

### RNA isolation and real time quantitative PCR (RT-qPCR)

Total RNA from bladder cancer tissues (size at 2 mm$^3$) or cells (number at $2 \times 10^6$) was obtained by using TRIzol reagent (Thermo Fisher Scientific). Each RNA (1 μg) was reversed transcribed to complementary DNA by utilizing a PrimeScript RT Reagent Kit with gDNA Eraser (Takara, Beijing, China), following the manufacturer's instructions. After that, qPCR was performed with a SYBR Premix ExTaq II kit (Takara) on an ABI 7900 Real Time PCR system (Applied Biosystems, Foster City, CA, USA) to quantify the relative the expression of *ROR1-AS1* and miR-504. The qPCR cycling conditions: one denaturation step of 10 min at 95°C; 40 cycles, with one cycle consisting of 10 sec at 95°C, 30 sec at 58°C, and 30 sec at 72°C. The primer sequences were shown in Table 1. The glyceraldehyde-3-phosphate dehydrogenase (*GAPDH*) and small nuclear RNA U6 (*U6*) genes were served as the internal controls. The $2^{-\Delta\Delta Ct}$ method was utilized to analyze the data and calculate the relative expression of each gene [22]. The experiments were performed in triplicate and repeated three times.

### Transfection

The specific short hairpin RNA (shRNA) targeting *ROR1-AS1* (shRNA-ROR1-AS1, 5′ -GUAGGGAAGUACAAUUUUUGAGUUA-3′) and the corresponding non-specific shRNA (shRNA-NC, 5′ -UGCAUCCUACUAGAUGGCCUGUAA-3′) were obtained from GenePharma Biotechnology (Suzhou, China). The miR-504 inhibitor (5′ -CAGACGUGAGAUAGACAUAA GAA-3′) and inhibitor NC (5′ -AUUCCCGAAUCCAAUAGUGCAU-3′) were synthesized in RiboBio Biotechnology (Guangzhou, China). T24 and 5637 cells were cultured in 6-well plates

**Table 1. The primer sequences used for real time quantitative PCR (RT-qPCR).**

| Gene symbol | Forward (5'-3') | Reverse (5'-3') |
|---|---|---|
| *ROR1-AS1* | GACGAAACACTGGAA | GTCTGATTTGGTAGCTT |
| *GAPDH* | CCAAAATCAAGTGGGGC | TGATGGCATGGACTGTGGT |
| miR-504 | GCTGCTGTTGGGAGACC | GCCCTCTGTATGGGAAAC |
| *U6* | CTCGCTTCGGCAGCACATA | ACGCTTCACGAATTTGCGT |

to get 80% confluence and then transfected with the indicated amounts of shRNA-ROR1-AS1 or shRNA-NC and miR-504 inhibitor or inhibitor NC by using lipofectamine 2000 transfection reagent (Thermo Fisher Scientific), according to the specific instructions. The transfection efficiency was analyzed after 48 h transfection by RT-qPCR.

## MTT assay

MTT (3-(4,5-dimethylthiazol-2yl)-2,5-diphenyl tetrazolium bromide) assay was applied to detect cell proliferation ability. T24 and 5637 cells were seeded into 96-well culture plate 24 h prior to transfection with shRNA-ROR1-AS1 or shRNA-NC and miR-504 inhibitor or inhibitor NC. After 0, 24, 48 or 72 h transfection, 12 μl of MTT solution (TIANGEN Biotechnology, Beijing, China) at 5 mg/ml concentration was added to each well, and incubated for 3 h at 37˚C. Then, the supernatants were discarded, and 120 μl of DMSO (TIANGEN Biotechnology) was used to solubilize the crystals for 20 min at 37˚C. Data of absorbance were measured at a wavelength of 480 nm (with 590 nm as the reference wavelength) using a Synergy™ HT Multi-Mode Microplate Reader (Biotek, Winooski, VT, USA) at the indicated time points.

## Wound scratch assay

Wound scratch assay was performed to evaluate cell migration capacity. Briefly, T24 and 5637 cells transfected with shRNA-ROR1-AS1 or shRNA-NC and miR-504 inhibitor or inhibitor NC were seeded into six-well culture plates and cultured with DMEM medium and 10% FBS to form a tight monolayer. Scraped lines were created with 100 μl sterile pipette tips, and the cell debris was removed with PBS and the remaining cells were incubated for 24 h at 37˚C with no serum-containing DMEM medium. The migrated distances of the growing edge on the monolayer were observed by using a ECLIPSE TS100 light microscope (Nikon Corporation, Tokyo, Japan) under a 200× microscope field at 0~24 h after being wounded.

## Luciferase reporter gene assay

A target prediction tool, starBase v2.0 (http://starbase.sysu.edu.cn/starbase2/), was used to predict potential miRs target of *ROR1-AS1*, and we found *ROR1-AS1* could potentially bind with miR-504 sequence. The *ROR1-AS1* transcript 1 containing the binding sites and non-binding sites for miR-504 sequence were synthesized and inserted into a psiCHECK2 reporter vector (Promega, WI, USA) to get the ROR1-AS1-WT and ROR1-AS1-MUT reporter vectors, respectively. Subsequently, bladder cancer cells were seeded into 24-well plates and transfected with the ROR1-AS1-WT and ROR1-AS1-MUT vector, together with miR-504 inhibitor or inhibitor NC using Lipofectamine 2000, according to the manufacturer's instructions. 24 h after the cell transfection, the luciferase activity was determined by using a dual-luciferase reporter assay (Promega, WI, USA). The relative Renilla luciferase activities were normalized to Firefly luciferase activities, which was served as an internal control for transfection efficiency.

## Statistical analysis

The statistical analyses were performed using SPSS 19.0 (SPSS Inc., Chicago, IL, USA). Data were presented as mean ± SD (standard deviation) from the three independent experiments. The expression differences between bladder cancer tissues and adjacent matched normal bladder tissues were analyzed using paired student's t-test. Chi-square test was used for correlation analysis. Kaplan-Meier method was utilized for analysis of prognosis. CCK-8 assay were analyzed using ANOVA analysis followed by post hoc testing. *P* values were two-sided and a two-tailed value of $p < 0.05$ was considered to be a statistically significant difference.

## Results

### *ROR1-AS1* expression is upregulated in patients with bladder cancer and associates with malignant clinicopatholigcal features

The relative expression levels of *ROR1-AS1* was determined by RT-qPCR in a total of 65 cases of bladder cancer patients. Compared with adjacent matched normal bladder tissues, the *ROR1-AS1* levels were notably upregulated in bladder cancer tissues (Fig 1A, $p < 0.05$). Subsequently, we analyzed the expression levels of *ROR1-AS1* in multiple bladder cancer cell lines (T24, 5637, J82, 253J and RT4). *ROR1-AS1* expression was also increased significantly in the five bladder cancer cells compared with the normal bladder epithelial cell (SV-HUC-1) (Fig 1B, $p < 0.05$). Furthermore, the correlations between *ROR1-AS1* expression and clinicopathologic variables was evaluated. The 65 pairs of bladder cancer patients were divided into two groups based on the median value of relative *ROR1-AS1* expression level: low (n = 33) and high (n = 32) *ROR1-AS1* expression groups. As shown in Table 2, the high *ROR1-AS1* expression was notably correlated with advanced tumor stage (p = 0.005), higher tumor grade (p = 0.045), and positive lymph node metastasis (p = 0.015). But not correlated with patient's sex (p = 0.097), age (p = 0.199), smoking status (p = 0.590) and tumor size (p = 0.505). Moreover, patients with high *ROR1-AS1* expression had a significantly shorter survival times

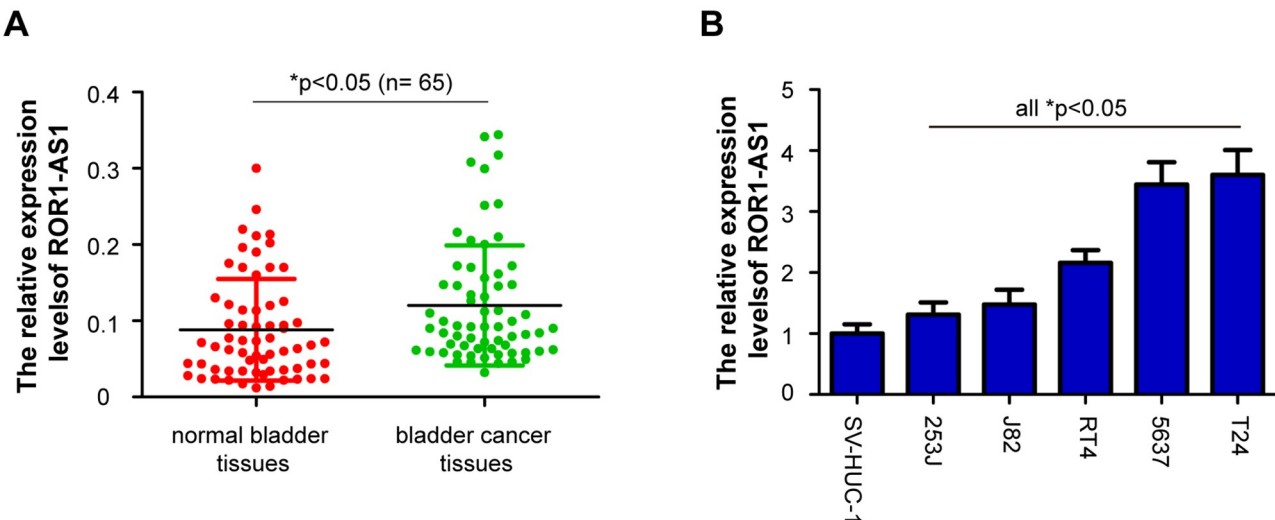

**Fig 1. The expression of lncRNA *ROR1-AS1* is upregulated in bladder cancer samples and cell lines.** (A) Real time quantitative PCR (RT-qPCR) analysis of ROR1 antisense RNA 1 (*ROR1-AS1*) expression in 65 cases of pairs bladder cancer tissues and adjacent matched normal bladder tissues. (B) *ROR1-AS1* expression was determined by RT-qPCR in five bladder cancer cell lines (T24, 5637, J82, 253J and RT4) and a normal bladder epithelial cell line (SV-HUC-1). Data were expressed as the mean ± SD (n = 3). *p < 0.05.

**Table 2. Correlation between ROR1 antisense RNA 1 (*ROR1-AS1*) expression and different clinicopathologic features in 65 cases of bladder cancer patients.**

| Clinicopathologic features | No. | ROR1-AS1 | | p |
| --- | --- | --- | --- | --- |
| | | High (n, %) | Low (n, %) | |
| Age (years) | | | | 0.199 |
| <55 | 19 | 7 (10.8) | 12 (18.5) | |
| ≥55 | 46 | 25 (38.5) | 21 (32.3) | |
| Sex | | | | 0.097 |
| Male | 38 | 22 (33.8) | 16 (24.6) | |
| Female | 27 | 10 (15.4) | 17 (26.2) | |
| Smoking status | | | | 0.590 |
| No | 14 | 6 (9.2) | 8 (12.3) | |
| Yes | 51 | 26 (40.0) | 25 (38.5) | |
| Size | | | | 0.505 |
| <3 cm | 25 | 11 (16.9) | 14 (21.5) | |
| ≥3 cm | 40 | 21 (32.3) | 19 (29.2) | |
| Grade | | | | 0.045* |
| G1-G2 | 22 | 7 (10.8) | 15 (23.1) | |
| G3 | 43 | 25 (38.5) | 18 (27.7) | |
| Stage | | | | 0.005* |
| T1-T2 | 21 | 5 (7.7) | 16 (24.6) | |
| T3 | 44 | 27 (41.5) | 17 (26.2) | |
| Lymph node metastasis | | | | 0.015* |
| No | 26 | 8 (12.3) | 18 (27.7) | |
| Yes | 39 | 24 (36.9) | 15 (23.1) | |

*$p < 0.05$.

compared with those patients with low *ROR1-AS1* expression (Fig 2, p<0.05). These data indicated that upregulation of *ROR1-AS1* may be involved in the progression of bladder cancer.

### *ROR1-AS1* knockdown suppresses the proliferation and migration of bladder cancer cells

To examine the potential roles of *ROR1-AS1* in the proliferation and migration of bladder cancer, T24 and 5637 cells were chose and treated with shRNA-ROR1-AS1 or shRNA-NC using MTT and wound scratch assays. As presented in Fig 3A, ROR1-AS1 siRNA transfected T24 cells significantly decreased *ROR1-AS1* expression levels, and knockdown of *ROR1-AS1* inhibited T24 cell growth and migration (p<0.05). Meanwhile, shRNA-ROR1-AS1 treated cells downregulated *ROR1-AS1* expression in 5637 cells, and arrested 5637 cell proliferation and migration (Fig 3B, p<0.05). These data demonstrated that *ROR1-AS1* contributes to cell proliferation and migration in bladder cancer.

### *ROR1-AS1* acts as a molecular sponge to decrease miR-504 expression

Using bioinformatics software starBase 2.0, we found that *ROR1-AS1* contained a binding sites for miR-504 sequence. RT-qPCR found that miR-504 was markedly downregulated in bladder cancer tissues than matched normal bladder tissues (Fig 4A, p<0.05). Pearson's correlation analysis showed that miR-504 expression was negatively correlated with *ROR1-AS1* expression in bladder cancer samples (Fig 4B, p<0.05). The potential binding sites for miR-504, ROR1-AS1-WT and ROR1-AS1-MUT were constructed in luciferase reporter gene vectors (Fig 4C).

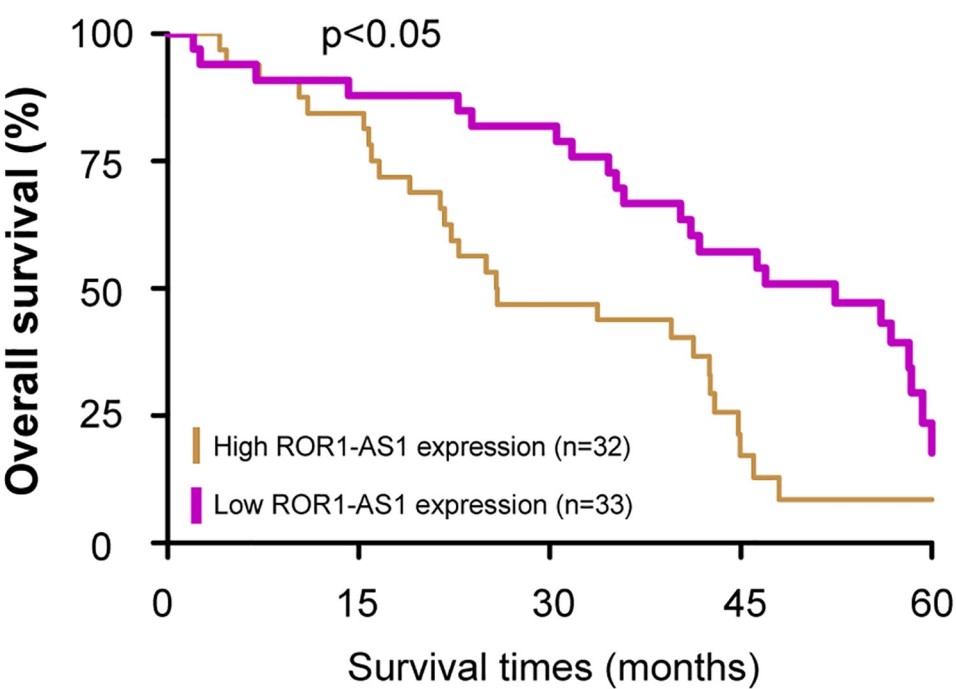

**Fig 2. Kaplan-Meier overall survival curves by *ROR1-AS1* expression.** Patients with high *ROR1-AS1* expression exhibited a significantly shorter survival times compared with those patients with low *ROR1-AS1* expression.

Results showed that transfection of miR-504 inhibitor in T24 and 5637 cells significantly suppressed miR-504 expression (Fig 4D, p<0.05). Knockdown of miR-504 increased the luciferase activity of ROR1-AS1-WT vector, but not ROR1-AS1-MUT vector in T24 and 5637 cells (Fig 4E, p<0.05). Transfection of shRNA-ROR1-AS1 into bladder cancer cells showed upregulation of miR-504 expression, but co-transfection with shRNA-ROR1-AS1 and miR-504 inhibitor reversed these effects (Fig 4F, p<0.05). These findings indicated that *ROR1-AS1* bind with miR-504 and acts as a molecular sponge to decrease miR-504 expression.

## Inhibition of miR-504 partly abrogates *ROR1-AS1* knockdown-induced inhibitory effects on bladder cancer cell growth and migration

Due to the negatively regulation of *ROR1-AS1* on miR-504 in bladder cancer cells, we speculated that the role of *ROR1-AS1* in regulating bladder cancer cell proliferation and migration was mediated by sponging miR-504 expression. We transfected miR-504 inhibitor or inhibitor NC into the shRNA-ROR1-AS1 treated T24 and 5637 cells, and functional rescue experiments were performed. Interesting, we found that *ROR1-AS1* knockdown mediated inhibitory effects on bladder cancer cell proliferation and migration were partially reversed by co-transfection with shRNA-ROR1-AS1 and miR-504 inhibitor (Fig 5A and 5B, p<0.05).

## Discussion

As an emerging hotspot in the investigation of human cancer, disregulation of lncRNAs in bladder cancer have been well studied. Some lncRNAs in bladder cancer can be utilized as potential therapeutic targets [23,24]. For example, Li et al [25] found that deleted in lympho-cytic leukemia 1 is upregulated in bladder cancer tissues and patients with high *DLEU1* expression exhibits a shorter survival time, and *DLEU1* increases cell proliferation, invasion, and

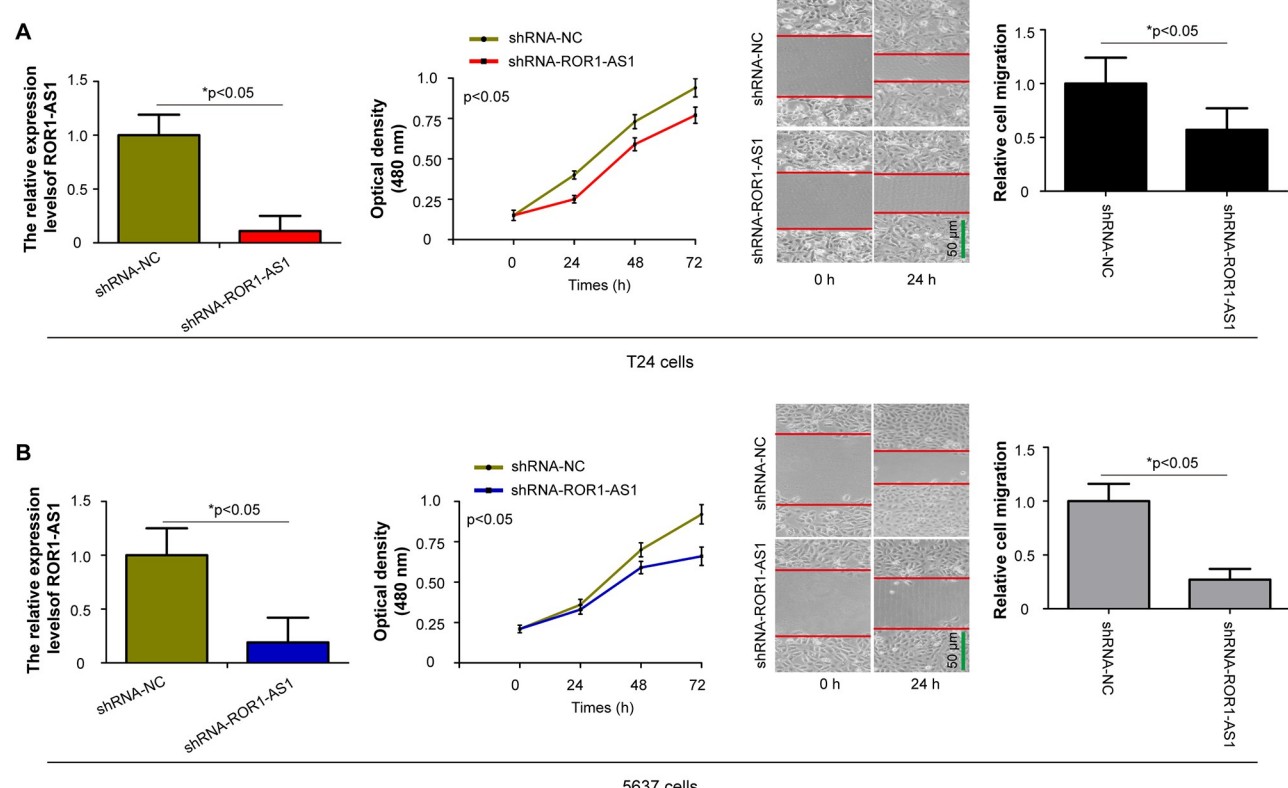

**Fig 3. The effects of *ROR1-AS1* knockdown on bladder cancer cells proliferation and migration.** (A) The T24 cells were transfected with shRNA-ROR1-AS1 or shRNA-NC, and *ROR1-AS1* expression levels were analyzed by RT-qPCR after 48 h transfection. 3-(4,5-dimethyl-2-thiazolyl)-2,5-diphenyl-2-H-tetrazolium bromide (MTT) and wound scratch assays were applied to determine T24 cells growth and migration *in vitro*. (B) shRNA-ROR1-AS1 treated cells downregulated *ROR1-AS1* expression in 5637 cells. *ROR1-AS1* knockdown obviously suppressed 5637 cell proliferation and migration. Data from at least three independent experiments, shown as the mean ± SD. $^*$p < 0.05.

cisplatin resistance of bladder cancer cells. Miao et al [26] showed that long intergenic non-protein coding RNA 612 facilitates the proliferation and invasion capacity of bladder cancer cells, suggesting that the lncRNA may act as a potential biomarker and therapeutic target. Gao et al [27] revealed that ZEB1-AS1/miR-200b/FSCN1 axis may serve as a potential target for molecular therapies of bladder cancer. Though there are many researches focusing on the regulation of lncRNAs on bladder cancer, there are still some potential mechanism need to be explored. To the best of our knowledge, this is the first report to illustrate the role of *ROR1-AS1* in bladder cancer. Moreover, we demonstrated that increased *ROR1-AS1* promotes the proliferation and migration of bladder cancer by sponging miR-504, and suggesting *ROR1-AS1* may be used as a prognostic biomarker and therapeutic target for bladder cancer treatment.

Previous studies have indicated that antisense lncRNAs including those located antisense to cancer-related genes were participated in complicate and accurate gene-net of human cancers [28]. *ROR1-AS1* is a novel antisense lncRNAs, which was first identified in mantle cell lymphoma, and could mediate tumor growth by histone modification through enhancer of zeste 2 polycomb repressive complex 2 subunit [17]. Recent studies have demonstrated that deregulation of *ROR1-AS1* contributes to carcinogenesis. In colorectal cancer, Wang et al [29] found that *ROR1-AS1* promotes cell metastasis and proliferation via inducing Wnt/β-catenin signaling pathway. However, the function and molecular mechanism of *ROR1-AS1* in bladder cancer

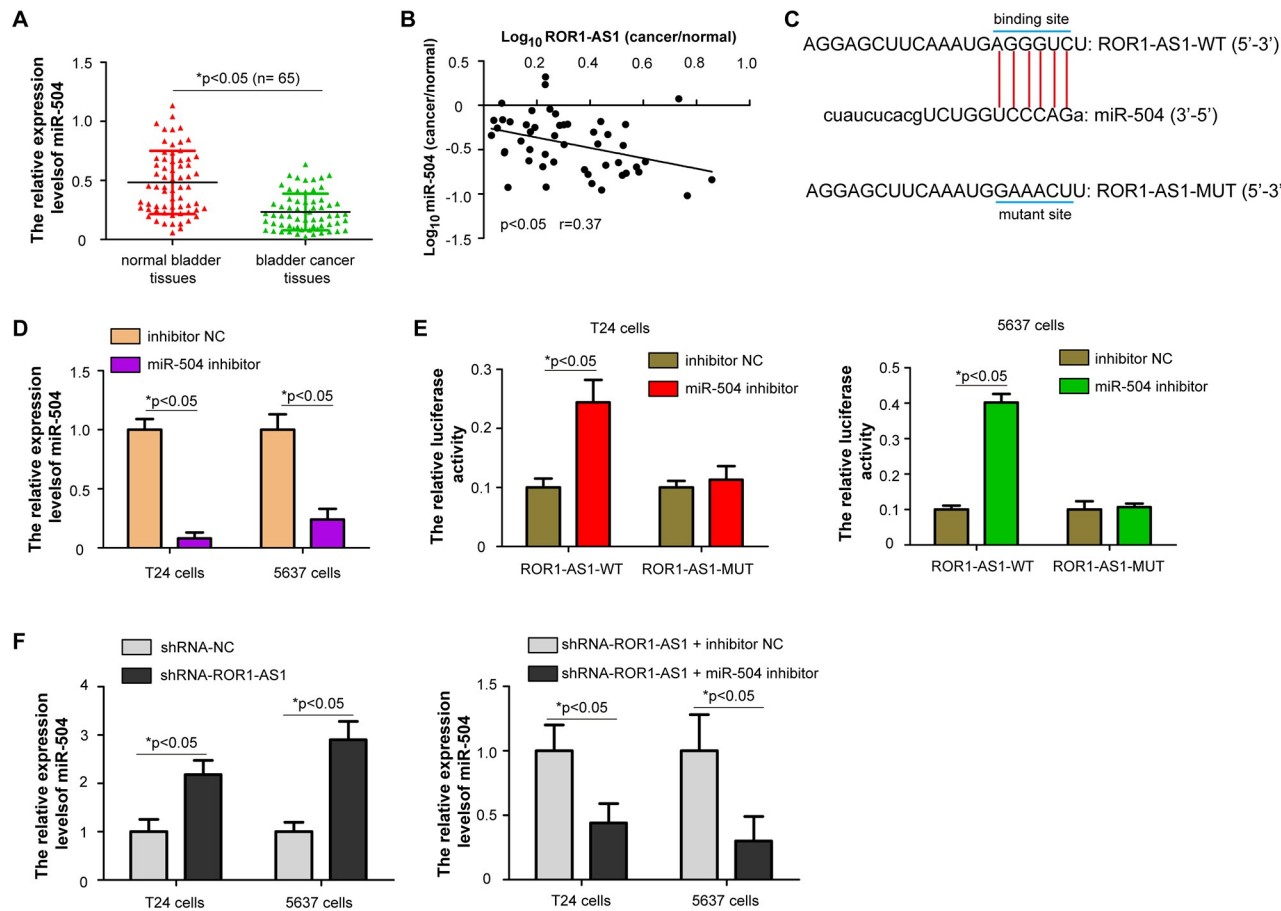

**Fig 4. The regulatory effects of *ROR1-AS1* on miR-504.** (A) RT-qPCR analysis of miR-504 expression in 65 cases of bladder cancer samples. (B) The correlation of miR-504 and *ROR1-AS1* expression in bladder cancer samples was evaluated by Pearson's correlation analysis. (C) Sequences of *ROR1-AS1* binding sites within the miR-504. (D) The miR-504 expression in T24 and 5637 cells was detected by RT-qPCR after transfection with miR-504 inhibitor or inhibitor NC. (E) Luciferase assay was measured 24 h post transfection in bladder cancer cells, which were co-transfected with ROR1-AS1-WT and ROR1-AS1-MUT vectors, together with miR-504 inhibitor or inhibitor NC. (F) Transfection of shRNA-ROR1-AS1 into bladder cancer cells showed upregulation of miR-504 expression, but co-transfection with shRNA-ROR1-AS1 and miR-504 inhibitor reversed these effects. Data were presented as the mean ± SD (n = 3). *p < 0.05.

are largely unknown until now. In this study, we identified that *ROR1-AS1* expression was increased in bladder cancer tissues and cell lines. Furthermore, the high *ROR1-AS1* expression levels were closely correlated with higher histological grade, advanced tumor stage, and positive lymph node metastasis in patients with bladder cancer, indicating that upregulation of *ROR1-AS1* represents an aggressive phenotypes of bladder cancer. More importantly, series of functional experiments further showed that knockdown of *ROR1-AS1* expression inhibited cell growth and migration of bladder cancer cells in vitro. These results are consistent with prior reports suggesting a oncogenic role of *ROR1-AS1* in the progression of bladder cancer.

A better understanding of the mechanisms underlying the pathogenesis of bladder cancer is key for improvement of anticancer therapy. Increasing evidence has exhibited that lncRNAs compete for miRNAs response elements (MREs) with the driver genes strongly relevant to tumor progression by acting as a ceRNA [30]. To further identify the potential mechanism of *ROR1-AS1* affects bladder cancer cell proliferation and migration, we predicted and chose miR-504 as a potential sponge of *ROR1-AS1* using bioinformatic analysis. Recently, investigations demonstrated the tumor-suppressive role of miR-504 in hypopharyngeal squamous cell

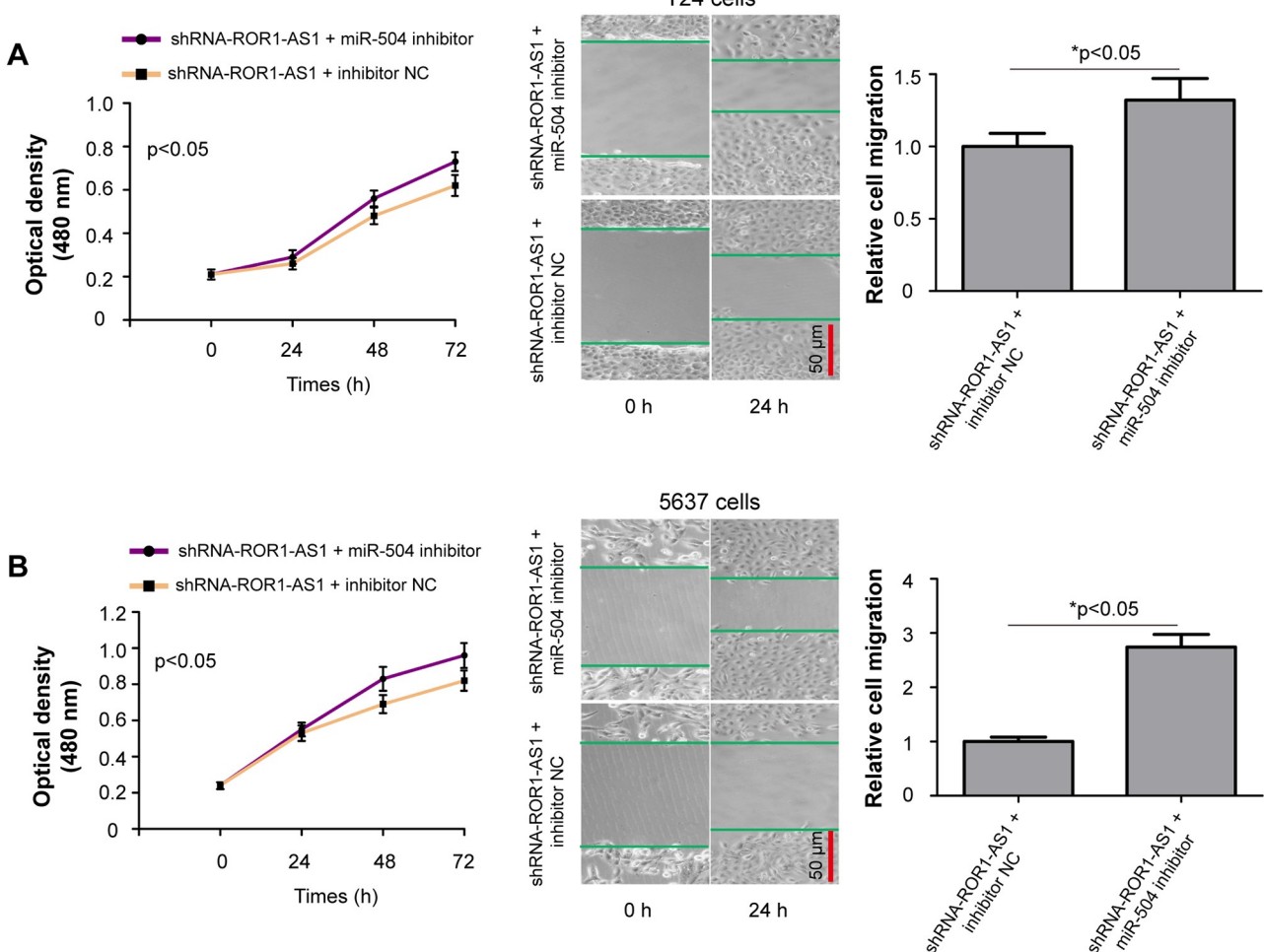

**Fig 5. *ROR1-AS1* promotes bladder cancer cells proliferation and migration by sponging miR-504.** (A) MTT and wound scratch assays were applied to determine T24 cells growth and migration *in vitro* after co-transfection with shRNA-ROR1-AS1 and miR-504 inhibitor or inhibitor NC. (B) Transfection of miR-504 inhibitor partly abrogated *ROR1-AS1* knockdown-induced inhibitory effects on 5637 cell growth and migration. Data were expressed as the mean ± SD (n = 3). *p < 0.05.

carcinoma and liver cancer [31,32]. Additional, Ye et al [33] showed that miR-504 is downregulated in non-small cell lung cancer tissues and can inhibit cell proliferation, invasion, and epithelial-mesenchymal transition by targeting lysyl oxidase like 2 expression. Similarly, Liu et al [34] found that miR-504 represses mesenchymal phenotype of glioblastoma by directly targeting the frizzled class receptor 7-mediated Wnt-β-catenin signaling pathway. However, role of miR-504 in bladder cancer is still unknown. In the present study, we first identified the relationship between miR-504 and *ROR1-AS1* in bladder cancer samples. Results revealed that miR-504 expression was downregulated and negatively correlated with *ROR1-AS1* expression. Moreover, luciferase reporter gene assay demonstrated that *ROR1-AS1* acts as a molecular sponge to decrease miR-504 expression. Functional rescue experiments confirmed that *ROR1-AS1* promotes cell growth and migration of bladder cancer via regulation of miR-504. Consistent with previous findings, we have validated that downregulated miR-504 acts as a tumor suppressor in bladder cancer.

In conclusion, our data suggested that upregulation of *ROR1-AS1* promotes bladder cancer cells proliferation and migration by regulating miR-504. Our findings contribute to a better

understanding of the importance of ROR1-AS1/miR-504 axis in bladder cancer progression, and provide a promising of lncRNA-based targeted approach for bladder cancer treatment. Further researches should also be conducted to illustrate the detailed molecular mechanism of *ROR1-AS1* in other urinary tumors.

## Author Contributions

**Conceptualization:** Qingke Chen.

**Formal analysis:** Lingmin Fu.

**Methodology:** Qingke Chen.

**Project administration:** Qingke Chen.

**Writing – original draft:** Qingke Chen, Lingmin Fu.

**Writing – review & editing:** Qingke Chen.

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
