## [Decision Letter · Decision Letter 0]

13 Dec 2019

PONE-D-19-31968

Upregulation of Long Non-Coding RNA ROR1-AS1 Promotes Cell Growth and Migration in Bladder Cancer by Regulation of MiR-504

PLOS ONE

Dear Mr Chen,

Thank you for submitting your manuscript to PLOS ONE. After careful consideration, we feel that it has merit but does not fully meet PLOS ONE’s publication criteria as it currently stands. Therefore, we invite you to submit a revised version of the manuscript that addresses the points raised during the review process.

We would appreciate receiving your revised manuscript by Jan 27 2020 11:59PM. To enhance the reproducibility of your results, we recommend that if applicable you deposit your laboratory protocols in protocols.io, where a protocol can be assigned its own identifier (DOI) such that it can be cited independently in the future. For instructions see: http://journals.plos.org/plosone/s/submission-guidelines#loc-laboratory-protocols

We look forward to receiving your revised manuscript.

Kind regards,

Wen-Jun Tu

Academic Editor

PLOS ONE

Journal Requirements:

2. We noticed you have some minor occurrence(s) of overlapping text with the following previous publication(s), which needs to be addressed:

https://doi.org/10.1016/j.canlet.2016.01.051

https://doi.org/10.18632/oncotarget.17956

In your revision ensure you cite all your sources (including your own works), and quote or rephrase any duplicated text outside the Methods section. Further consideration is dependent on these concerns being addressed.

3. In the ethics statement in the manuscript and in the online submission form, please provide additional information about the patient samples used in your retrospective study, including the date range (month and year) during which patients' samples were accessed by the authors.

Reviewers' comments:

Reviewer's Responses to Questions

**Comments to the Author**

1. Is the manuscript technically sound, and do the data support the conclusions?

Reviewer #1: Yes

Reviewer #2: Yes

2. Has the statistical analysis been performed appropriately and rigorously? 

Reviewer #1: Yes

Reviewer #2: Yes

3. Have the authors made all data underlying the findings in their manuscript fully available?

Reviewer #1: Yes

Reviewer #2: Yes

4. Is the manuscript presented in an intelligible fashion and written in standard English?

Reviewer #1: Yes

Reviewer #2: Yes

5. Review Comments to the Author

Reviewer #1: Whether the RT-qPCR experiment was repeated 3 times? the author needs to indicate in the method.

2.The sequences of shRNA-ROR1-AS1, shRNA-NC, miR-504 inhibitor and inhibitor NC should be added in the method.

3.The reason for choosing T24 and 5637 for functional assays, why not used J82, 253J and RT4

4.The scale bar should added in the images of wound scratch assay.

5.In discussion, they should discuss the role of ROR1-AS1 in other cancers and summarize them.

6.LncRNAs is very basic and information about the classification criteria is lacking. Therefore, it is suggested to mention and to add the corresponding references.

7.Please specifiy de size range of tissues used for RNA extraction.

8.There are some grammatical and wording errors in the paper. It is recommended that professionals revise it.

Reviewer #2: Increasing evidences showed that multiple long non-coding RNAs (lncRNAs) act crucial regulatory functions in the pathogenesis of bladder cancer. This study aimed to determine the expression and clinical significance of ROR1 antisense RNA (ROR1-AS1) from patients with bladder cancer, and to identify the potential role and mechanism underlying ROR1-AS1-related cancer progression. The results showed that increased ROR1-AS1 promotes cell growth and migration of bladder cancer via regulation of miR-504, and ROR1-AS1 may be used as a prognostic biomarker and therapeutic target for bladder cancer.

This study is interesting and clinically relevant, however there are a number of grammatical errors and issues regarding data collection that should be addressed.

Comments

1. The grammar needs to be edited throughout.

2. The introduction section, the aim of this study and more detailed research background should be presented.

3. “This research was approved by the Ethics Committee of the First Affiliated Hospital of Nanchang University’ the approval no should be present.

4. How many patients and bladder tissues were collected?

5. Human bladder tissue and tumor-adjacent normal bladder tissues were matched? Matched what?

6. Discussion: should be focused on the own results, It is written general and seems to be vague.

7. Kaplan-Meier survival analysis was used in the results section, please showed it in the method section.

8. Figure 2. Comments and headings should be in the same direction.

6. PLOS authors have the option to publish the peer review history of their article (what does this mean?). If published, this will include your full peer review and any attached files.

Reviewer #1: No

Reviewer #2: No

---

## [Author Response · Author response to Decision Letter 0]

16 Dec 2019

Dear editor,

Thank you very much for your attention and the referees’ valuable comments on our manuscript. We have revised the manuscript according to reviewers' comments. Enclosed please find the revised manuscript, responses to the referees as well as a list of changes. We sincerely hope this manuscript will be finally acceptable to be published on PLOS ONE. We look forward to hearing from you soon.

Best regard!

Yours Sincerely,

Qingke Chen, 

Department of Urology, First Affiliated Hospital of Nanchang University, No. 17, Yongwaizheng Street, Nanchang 330006, Jiangxi Province, China. Tel.: +86-0791-88692526, 

E-mail: cqkurethral@126.com.

Response: The revised manuscript has met PLOS ONE's style requirements.

2. We noticed you have some minor occurrence(s) of overlapping text with the following previous publication(s), which needs to be addressed:

https://doi.org/10.1016/j.canlet.2016.01.051

https://doi.org/10.18632/oncotarget.17956

In your revision ensure you cite all your sources (including your own works), and quote or rephrase any duplicated text outside the Methods section. Further consideration is dependent on these concerns being addressed.

Response: We have quoted and/or rephrased any duplicated texts of the above two article in the revised manuscript.

3. In the ethics statement in the manuscript and in the online submission form, please provide additional information about the patient samples used in your retrospective study, including the date range (month and year) during which patients' samples were accessed by the authors.

Response: The date range in which human subjects’ data/samples were collected and the date(s) conducted this study have been added in the revised manuscript and online submission form.

Comments to the Author

Reviewer #1:

1: Whether the RT-qPCR experiment was repeated 3 times? the author needs to indicate in the method.

Response: Thanks for your comment. The repeated 3 times for RT-qPCR has been added in the revised manuscript. Please see it in page 7, line 144-145.

2.The sequences of shRNA-ROR1-AS1, shRNA-NC, miR-504 inhibitor and inhibitor NC should be added in the method.

Response: The sequences of shRNA-ROR1-AS1, shRNA-NC, miR-504 inhibitor and inhibitor NC have been added in the revised manuscript. Please see it in page 7, line 149-153.

3.The reason for choosing T24 and 5637 for functional assays, why not used J82, 253J and RT4.

Response: The expression levels of ROR1-AS1 in T24 and 5637 cells were higher than that in the 253J and RT4 cells, and were chosen for further functional experiments.

4.The scale bar should added in the images of wound scratch assay.

Response: The scale bars have been added in the revised Fig. 3 and Fig. 5.

5.In discussion, they should discuss the role of ROR1-AS1 in other cancers and summarize them.

Response: Thanks for your kind advice. Just like your suggestion, the roles of ROR1-AS1 in colorectal cancer and mantle cell lymphoma have been summarized and added in the Discussion. Please see it in page 15, line 316-322.

6.LncRNAs is very basic and information about the classification criteria is lacking. Therefore, it is suggested to mention and to add the corresponding references.

Response: The classification criteria for lncRNAs and the corresponding references have been added in Introduction. Please see it in page 4, line 72-74.

7.Please specifiy de size range of tissues used for RNA extraction.

Response: Tissue RNA was extracted from tissues (size at 2 mm3) or cells (number at 2 × 106). Please see it in page 6, line 132.

8.There are some grammatical and wording errors in the paper. It is recommended that professionals revise it.

Response: We sorry for the these errors in the original manuscript. The errors have been corrected with assistance from Elsevier Language Editing Services with appropriate research background.

Reviewer #2: Increasing evidences showed that multiple long non-coding RNAs (lncRNAs) act crucial regulatory functions in the pathogenesis of bladder cancer. This study aimed to determine the expression and clinical significance of ROR1 antisense RNA (ROR1-AS1) from patients with bladder cancer, and to identify the potential role and mechanism underlying ROR1-AS1-related cancer progression. The results showed that increased ROR1-AS1 promotes cell growth and migration of bladder cancer via regulation of miR-504, and ROR1-AS1 may be used as a prognostic biomarker and therapeutic target for bladder cancer.

This study is interesting and clinically relevant, however there are a number of grammatical errors and issues regarding data collection that should be addressed.

Comments

1. The grammar needs to be edited throughout.

Response: Thanks for your comment. We sorry for the these errors in the original manuscript. The errors have been corrected with assistance from Elsevier Language Editing Services with appropriate research background.

2. The introduction section, the aim of this study and more detailed research background should be presented.

Response: The aim of this study and detailed research background have been added in the revised manuscript. Please see it in page 5, line 96-104.

3. “This research was approved by the Ethics Committee of the First Affiliated Hospital of Nanchang University’ the approval no should be present.

Response: The approval no. (Approval No. 2018070) has been added in the revised manuscript. Please see it in page 5, line 109.

4. How many patients and bladder tissues were collected?

Response: This retrospective study included 65 cases of bladder cancer patients who underwent surgery at Department of Urology, First Affiliated Hospital of Nanchang University between 09/2011 and 05/2017. 65 human bladder tissue and tumor-adjacent normal bladder tissues from all patients were collected between 09/2011 and 05/2017. Please see it in page 6, line 111-122.

5. Human bladder tissue and tumor-adjacent normal bladder tissues were matched? Matched what?

Response: Yes. Tumor tissues matched normal tissues. 

6. Discussion: should be focused on the own results, It is written general and seems to be vague.

Response: This is an excellent advice. The Discussion has been revised according the comment. Please see it in Discussion.

7. Kaplan-Meier survival analysis was used in the results section, please showed it in the method section.

Response: Kaplan-Meier survival analysis has been deleted in the Results section, and added in the Method section. Please see it in page 10, line 205-206.

8. Figure 2. Comments and headings should be in the same direction.

Response: The Fig.2 has been revised according the suggestion.

---

## [Decision Letter · Decision Letter 1]

23 Dec 2019

Upregulation of Long Non-Coding RNA ROR1-AS1 Promotes Cell Growth and Migration in Bladder Cancer by Regulation of MiR-504

PONE-D-19-31968R1

Dear Dr. Chen,

We are pleased to inform you that your manuscript has been judged scientifically suitable for publication and will be formally accepted for publication once it complies with all outstanding technical requirements.

With kind regards,

Wen-Jun Tu

Academic Editor

PLOS ONE

Additional Editor Comments (optional):

Reviewers' comments:

Reviewer's Responses to Questions

**Comments to the Author**

1. If the authors have adequately addressed your comments raised in a previous round of review and you feel that this manuscript is now acceptable for publication, you may indicate that here to bypass the “Comments to the Author” section, enter your conflict of interest statement in the “Confidential to Editor” section, and submit your "Accept" recommendation.

Reviewer #1: All comments have been addressed

Reviewer #2: All comments have been addressed

2. Is the manuscript technically sound, and do the data support the conclusions?

Reviewer #1: Yes

Reviewer #2: Yes

3. Has the statistical analysis been performed appropriately and rigorously? 

Reviewer #1: Yes

Reviewer #2: Yes

4. Have the authors made all data underlying the findings in their manuscript fully available?

Reviewer #1: Yes

Reviewer #2: Yes

5. Is the manuscript presented in an intelligible fashion and written in standard English?

Reviewer #1: Yes

Reviewer #2: Yes

6. Review Comments to the Author

Reviewer #1: The problems raised in a previous round were reviewed and corrected by the authors carefully. And the English language has be revised and corrected . As a result, I think the current revised manuscript is suitable for publication.

Reviewer #2: The Authors had thoroughly revised the manuscript and addressed requested changes and improvements。 No further comments. Thanks

7. PLOS authors have the option to publish the peer review history of their article (what does this mean?). If published, this will include your full peer review and any attached files.

Reviewer #1: No

Reviewer #2: No

---

## [Editor Report · Acceptance letter]

30 Dec 2019

PONE-D-19-31968R1 

Upregulation of Long Non-Coding RNA ROR1-AS1 Promotes Cell Growth and Migration in Bladder Cancer by Regulation of MiR-504 

Dear Dr. Chen:

I am pleased to inform you that your manuscript has been deemed suitable for publication in PLOS ONE. Congratulations! Your manuscript is now with our production department. 

With kind regards,

on behalf of

Dr. Wen-Jun Tu 

Academic Editor

PLOS ONE